# Lower Strength Alcohol Products—A Realist Review-Based Road Map for European Policy Making

**DOI:** 10.3390/nu14183779

**Published:** 2022-09-13

**Authors:** Peter Anderson, Daša Kokole, Eva Jané Llopis, Robyn Burton, Dirk W. Lachenmeier

**Affiliations:** 1Department of Health Promotion, CAPHRI Care and Public Health Research Institute, Maastricht University, 6200 MD Maastricht, The Netherlands; 2Population Health Sciences Institute, Newcastle University, Newcastle upon Tyne NE2 4AX, UK; 3ESADE Business School, Ramon Llull University, 08034 Barcelona, Spain; 4Institute of Psychiatry, Psychology, and Neuroscience, King’s College London, London SE5 8AF, UK; 5Chemisches und Veterinäruntersuchungsamt (CVUA) Karlsruhe, Weissenburger Straße 3, 76187 Karlsruhe, Germany

**Keywords:** realist review, lower strength alcohol products, substitution, household purchase data

## Abstract

This paper reports the result of a realist review based on a theory of change that substitution of higher strength alcohol products with lower strength alcohol products leads to decreases in overall levels of alcohol consumption in populations and consumer groups. The paper summarizes the results of 128 publications across twelve different themes. European consumers are increasingly buying and drinking lower strength alcohol products over time, with some two fifths doing so to drink less alcohol. It tends to be younger more socially advantaged men, and existing heavier buyers and drinkers of alcohol, who take up lower strength alcohol products. Substitution leads to a lower number of grams of alcohol bought and drunk. Although based on limited studies, buying and drinking lower strength products do not appear to act as gateways to buying and drinking higher strength products. Producer companies are increasing the availability of lower strength alcohol products, particularly for beer, with extra costs of production offset by income from sales. Lower strength alcohol products tend to be marketed as compliments to, rather than substitutes of, existing alcohol consumption, with, to date, the impact of such marketing not evaluated. Production of lower strength alcohol products could impair the impact of existing alcohol policy through alibi marketing (using the brand of lower strength products to promote higher strength products), broadened normalization of drinking cultures, and pressure to weaken policies. In addition to increasing the availability of lower strength products and improved labelling, the key policy that favours substitution of higher strength alcohol products with lower strength products is an alcohol tax based on the dose of alcohol across all products.

## 1. Introduction

The alcohol in alcoholic beverages is toxic to many bodily systems [1,2,3,4,5], is genotoxic and is a carcinogen [6,7], being a cause of cancers of the oral cavity, pharynx, larynx, oesophagus, colorectum, liver (hepatocellular carcinoma) and female breast [8,9,10]. Within the European Union (EU), a population 447 million people as of 2020, alcohol is responsible for a little under 300,000 deaths a year, with cancer the top cause of alcohol-related deaths (29% of all deaths due to alcohol), followed by liver cirrhosis (20% of all deaths due to alcohol), and cardiovascular diseases (19% of all deaths due to alcohol) (data for 2016) [11]. Alcohol is also detrimental to societal well being, with societal costs ranging from 0.12% to 3.47% of GDP across EU countries [12,13].

Better health and well being are at the heart of the United Nations Agenda 2030, which has an overall aim to drive transformative change to shift the world onto a sustainable and resilient path through cross-sectoral and cross-cutting actions [14]. Target 3.5 of Agenda 2030 is to “strengthen the prevention and treatment of substance abuse, including narcotic drug abuse and harmful use of alcohol” [14], operationalized as a non-communicable disease (NCD) target by the World Health Organization (WHO) to reduce the harmful use of alcohol in relative terms by 10% between 2010 and 2025 [15].

Promoting sustainable health, the key to reducing the harm done by alcohol, is to drink less alcohol [16]. The WHO SAFER initiative calls on governments to encourage people to drink less alcohol by making alcohol more expensive, decreasing its availability, banning or strictly regulating its advertising, putting in place strict drink-driving laws, and providing advice, support and treatment to reduce consumption [17]. Alcohol policies in general, including setting a minimum price per gram of alcohol sold, and regulating or banning advertising generate savings in heath expenditure, and improve employment and productivity, with high returns on investment [18].

In its action plan (2022–2030) to effectively implement the global strategy to reduce the harmful use of alcohol as a public health priority, in addition to the “continued enforcement of high-impact cost-effective policy options included in the WHO SAFER initiative” [17], WHO calls on economic operators to “substitute, whenever possible, higher-alcohol products with no-alcohol and lower-alcohol products in their overall product portfolios, with the goal of decreasing the overall levels of alcohol consumption in populations and consumer groups, while avoiding the circumvention of existing regulations for alcoholic beverages and the targeting of new consumer groups with alcohol marketing, advertising and promotional activities” [19].

Driven by the proposals of the WHO action plan, this paper reports a realist review [20,21], a method of review of complex policy interventions, to gain better perspectives on lower strength alcohol products and their potential public health impact. For this purpose, the paper surveys the evidence on the hypothesized theory of change that substitution of higher strength alcohol products with no-alcohol and lower alcohol products leads to decreases in overall levels of alcohol consumption in populations and consumer groups [22], with the aim to better understand what it is about lower strength alcohol products that could reduce risk to ill health, for whom and in what circumstances.

At the outset, it is important to note that different countries use different terms, with both non-alcoholic and alcohol-free referring to the same products; in addition, the alcohol-free descriptor varies across countries, with, for example, the definition of alcohol-free ranging from an alcohol by volume, ABV, of 0.05% to ≤1.2%. By lower strength products, this review includes fully de-alcoholised products (such as beers and wines with an alcohol by volume, ABV,= 0.0%), no-alcohol products (such as beers and wines with an ABV = 0.5%) [23], reformulated existing products to include less alcohol (such as beers, whose ABV is reduced), and the production of new lower strength products (such as variants of whiskey and gin with an ABV = 20%).

Based on Medical Research Council’s guidance [24], and informed by a previous scoping review [25], a twelve-component logic model is adopted (Figure 1).

Based on the findings, a road map is proposed for substituting higher strength alcohol products with lower strength products, alerting policymakers to the problems that might arise and how to deal with them.

## 2. Methods

The realist review was undertaken following the guidance of Pawson and colleagues [20] and of Rycroft-Malone and colleagues [21]. The starting point is a consideration that health gain can be achieved from substitution of higher strength alcohol products with lower strength products [22], as proposed by WHO [19]. To examine each of the twelve components of the logic model, we undertook targeted and broad-ranged multiple searches identifying both scientific publications and European-based grey literature [26], with Europe being the focus of our policy-based road map. We used multiple search strategies, retrieving materials purposively to address the specific components, and continuing searching until the retrieved literature did not add anything new to our understanding of the components and when we considered that further searching would be unlikely to add new knowledge. Given the broad range of questions to be answered, and the need to capture grey literature [26], we based our search on both Google and Google Scholar, searching for publications in the English language (or, with English-language summaries) from 1 January 2011 until 31 May 2022 that were not identified in a previous scoping review [25]. Our search approach was iterative and interactive, tracking back and forth from the literature retrieved to the twelve components, with the search terms used evolving in the process. We also used snowballing, pursuing references of publications by hand.

## 3. Results

For each component of Figure 1, results are presented in narrative form, first for grey literature, and, second, for academic publications. Table 1 summarizes the key findings qualitatively by component, with the associated publication numbers, available in the reference list.

### 3.1. Implementation

#### 3.1.1. What Is the Extent of the Production of Lower Strength Products?

##### Grey Literature

One grey-literature publication found: among 10 focus markets examined in the 2022 IWSR No- and Low-Alcohol Strategic Study (Australia, Brazil, Canada, France, Germany, Japan, South Africa, Spain, the United Kingdom, and the United States), the market value of no/low alcohol products in 2021 was estimated at just under USD 10 billion, up from USD 7.8bn in 2018 [27]. The report forecasted that no- and low-alcohol volume would grow by +8% compound annual growth rate (CAGR) between 2021 and 2025, compared to regular alcohol volume growth of +0.7% CAGR during that same period. No-alcohol beer was projected to grow at +11% CAGR between 2021–2025, and no-alcohol ready-to-drinks (RTDs) and no-alcohol spirits at +14% CAGR. Wine was expected to differ, with low-alcohol wine expected to grow at +20%, and no-alcohol wine at +9% CAGR.

##### Academic Publications

Routinely published Eurostat Data is only available for no-alcohol beers, and not for other products; further, data is not available for all EU countries. An analysis of Eurostat Data from 18 out of 27 EU countries found that sold production of beer with an ABV ≤ 0.5% represented 3.8% of the volume of sold production of all beer in 2019, having increased from 1.8% in 2013 [28]. Five countries accounted for 81% of sold production volume: Germany, the Netherlands, Spain, Poland and Czechia.

Based on British household purchase data, out of 1905 different beer brands purchased, 46 were low- and no-alcohol beer brands (with an ABV of 3.5% or less) and newly introduced during 2015 to 2018, with 41 having less than an average of one purchase a day across all households (64,280 households providing data) and one product responsible for 64% of all purchased millilitres of such beer [29]. During 2015 to 2018, 33 existing beer brands were reformulated to contain less alcohol, with 24 having less than an average of one purchase a day across all households, and one product whose ABV was reduced from 4.8% to 4.5% responsible for 71% of all post-reformulation purchased millilitres of such beer. During 2018, the volume of purchased new low- and no-alcohol beer products was 2.6% and the volume of reformulated beer products was 6.9% of the volume of all beer products purchased [29].

#### 3.1.2. To What Extent Are Consumers Buying and Drinking Lower Strength Alcohol Products?

##### Grey Literature

Three grey-literature publications found: (i) in 2018, 13% of Dutch adults reported drinking no-alcohol beer on a monthly basis [44]; (ii) in 2021, 20.3% of German adults reported having drunk no-alcohol beer, with 8.9% of respondents doing so at least monthly [45]; and, (iii) in 2021, 21% of British adults reported consuming a no-alcohol product during the past 12 months and 17% a low-alcohol product during the last 12 months [46].

##### Academic Publications

Based on PRODCOM data, the top four countries with the highest apparent consumption of no-alcohol beer as a percentage of total beer consumption are Czechia, the Netherlands, Spain and Luxembourg [28], with three of these countries having relatively high beer consumption (Czechia, the Netherlands, and Luxembourg) [141], which could explain the higher proportion of non-alcoholic beer as percentage of all beer in these countries. Consumption of no-alcohol beer does not seem to be related to changes in per-capita alcohol consumption; between 2010 and 2019, there was no change in per-capita alcohol consumption in Czechia (1% increase), a decrease in the Netherlands (6%), a large increase in Spain (25%), and no change in Luxembourg (1% decrease) [141].

Based on British data, household purchases for the years 2015 to 2019 found that 2.1% of all beer purchases had an alcohol by volume (ABV) ≤ 0.5% (1.9% with an ABV ≤ 0.05%), with small increases over time; 0.3% of all purchases of wine products had an ABV ≤ 0.5% (0.1% with an ABV ≤ 0.05%), with small increases over time; there were no purchases of products that emulate spirits with an ABV ≤ 0.5%. [30] Over the same time period, the average ABV of purchased beer decreased, whereas that of wines and spirits increased. For each day that a household made a purchase of beer, for every 100 mL of beer purchased, 93.6 mL was of higher strength beer (ABV > 3.5%), 4.9 mL was of lower strength beer (ABV > 0.05% and ≤3.5%) and 1.5 mL was of alcohol-free beer (ABV ≤ 0.05%) [30].

Based on Spanish data, household purchases for the years 2017 (from 2nd quarter) to 2022 (end of 1st quarter) found that 12.4% of all beer purchases and 3.9% of all wine purchases had an ABV ≤ 0.5%, with the trends for no-alcohol beer stable over time, and the trends for no-alcohol wine decreasing very slightly overtime [31].

#### 3.1.3. What Are the Currently Implemented Policies Relevant for Lower Strength Alcohol Products?

##### Grey Literature

On the whole, it seems that little policy is set that might favour substitution. Four grey-literature publications found or stated: (i) in 2020, the voluntary Dutch Advertising Code for Alcohol Free and Low Alcohol Beer came into effect, stipulating that advertising for alcohol-free and low-alcohol beer may not be aimed at young people under the age of 18, and that advertising for low-alcohol beer may not be aimed at pregnant women and drivers [32]. The realist review found no evaluation of the code. Based on previous experiences of evaluation of voluntary advertising codes [33], it is unlikely to have any favourable impact; (ii) in its consultation document, “Advancing our health: prevention in the 2020s”, whilst not without its critics [34], the UK Government made a commitment to work with the drinks industry to “deliver a significant increase in the availability of alcohol-free and low-alcohol products by 2025” [35]. The realist review found no concrete proposals for implementing the commitment, other than producing definitions of descriptors for lower strength products [36]; (iii) at the European level, Directive 92/83/EEC [37] on the structures of excise duty on alcohol and alcoholic beverages, which sets out the common rules on the structures of excise duty applied to alcohol and alcoholic beverages, has been amended (Directive (EU) 2020/1151) [38]. Member states may apply reduced rates of excise duty, which may fall below the minimum rate, for beer with an actual alcoholic strength by volume not exceeding 3.5% vol (previously, it was 2.8%). Member states that apply a duty for wine may apply reduced rates of excise duty for wine with an actual alcoholic strength by volume not exceeding 8.5% vol [37]. The impact of the amendment has not been evaluated, but for beer, at least in Spain, where it has been studied, only an extra 3.2% of low-alcohol beer sales would qualify for a reduced rate of excise duty [31], but in Great Britain (no longer an EU member state), an extra 36% of low-alcohol beer sales would qualify [30]; (iv) the Common Agriculture Policy was reformed at the end of 2021 (see Regulation (EU) 2021/2117), allowing wines with protected designation of origin (PDO) and protected geographical indication (PGI) to be partially de-alcoholised (i.e., down to an ABV > 0.5%) and marketed as such [39,40]; all non-PDO and non-PGI wines can be produced and marketed as de-alcoholised wines (i.e., down to an ABV of 0.0%). Such a reform could facilitate increased production of low-alcohol wines; this could have knock-on effects on increased production of zero- and no-alcohol non-PDO and non-PGI wines, although this may be beyond the capacity of most relatively small wine producers that would need to off source the de-alcoholization of their products [18]. The impact of the reform has not been evaluated.

On the other hand, it seems that much policy is set that disfavours substitution of higher strength products with lower strength products. Four grey-literature publications found or stated: (i) market research data from the United Kingdom, undertaken during 2020, found that 89% of respondents could not accurately define a low-alcohol drink (ABV ≤ 1.2%) and 80% could not accurately define an alcohol-free (ABV ≤ 0.05%) drink [46]. Of respondents, 62% agreed that the definitions of low-alcohol, de-alcoholised and no-alcohol drinks were confusing; (ii) Regulation (EU) 1169/2011 specifies that alcoholic products with an ABV ≤ 1.2% are not required to specify the ABV on the label [41]; (iii) Regulation (EC) No 1924/2006 states that “beverages containing more than 1.2% by volume of alcohol shall not bear: (a) health claims; (b) nutrition claims, other than those which refer to a reduction in the alcohol or energy content. In the absence of specific Community rules regarding nutrition claims referring to the reduction or absence of alcohol or energy in beverages which normally contain alcohol, relevant national rules may apply in compliance with the provisions of the Treaty” [42]. Thus, alcoholic beverages with an ABV ≤ 1.2% can bear health and nutrition claims within the specifications of the regulation; and, (iv) setting taxes based per ABV is restricted by Directive 92/84/EEC on the approximation of the rates of excise duty on alcohol and alcoholic beverages that sets out the minimum rates of excise duty on alcohol products [43]. Whereas, for beer, the minimum rate is set at 1.87 European Currency Units (ECU) per 100 litres (1.0 hectolitres) of beer per ABV of beer, for wine, the minimum rate is set at zero, and, for spirits (with a minimum ABV of 15%), the minimum rate is set at 550 ECU per 100 litres. Thus, the only product with excise duty related to ABV is beer [37]. Articles 9(1), 13(1), 18(1) and 21 in Directive 92/83/EEC would all require amendment to allow excise duty to be levied based on the ABV of the product [43].

##### Academic Publications

None.

### 3.2. Context

#### 3.2.1. Who Buys and Drinks Lower Strength Alcohol Products and Why?

##### Grey Literature

Several grey-literature publications report on surveys of who buys and drinks lower strength products; however, existing non-drinkers who take up lower strength products are not always captured by these surveys, with the consequence that the surveys are more likely to identify substitution rather than addition. In terms of who buys and drinks lower strength products, four grey-literature publications found: (i) in 2018 in the Netherlands, drinking non-alcoholic beer was more common amongst men, younger adults and those with higher levels of education [44]; (ii) in 2021, in Germany, men, the middle-aged and individuals with higher incomes were more likely to consume non-alcoholic beer [45]; (iii) in 2021, in the United Kingdom, drinking no- and low- alcohol products was more common amongst men, young adults, existing drinkers and those with higher incomes [46]; and, (iv) based on an the 2022 IWSR No- and Low-Alcohol Strategic Study (Australia, Brazil, Canada, France, Germany, Japan, South Africa, Spain, the United Kingdom, and the United States), young adults and higher income consumers were more likely to report the use of no- and low-alcohol products [27].

In terms of why consumers buy and drink lower strength products, three grey-literature publications found: (i) in a 2021 survey of Dutch adults that consume beer at least monthly, undertaken by the Dutch Brewers, the highest rated reasons for drinking no- or low-alcohol beer reported were: liking it (56%, increased from 37% in 2018); having to drive (47%, similar to 51% in 2018); and, wanting to drink less alcohol (44%, increased from 32% in 2018) [47]; (ii) data from the United Kingdom, collected during 2021, found that two fifths of respondents who had drunk no- and low- alcohol products within the previous 12 months did so because they were trying to drink less alcohol, with 7% doing so because they were recovering from “alcohol dependency” [46]; and, (iii) based on the IWSR market research report [27], among adults who had purchased no- and low-alcohol products, 37% of people reported that the reason for doing so was to avoid the effects of drinking alcohol, with 17% reporting that they were drinking no/low to avoid alcohol completely; a third of drinkers reported that they bought no/low alcohol because they enjoyed the taste [27].

##### Academic Publications

In terms of who buys and drinks lower strength products, household purchase data from Great Britain for the years 2015 to 2020 and market-research-based consumer surveys from Great Britain for the years 2015 to 2018 found that alcohol-free beer was more likely to be bought and drunk by those who generally bought and drank the most alcohol, those who bought and drank beer with an ABV > 3.5%, men, those with younger ages, and those with higher incomes and higher social grades [48], with gaps in buying alcohol-free beer between households in higher and lower social grades widening between 2015 and 2020 [48].

In terms of why consumers buy and drink lower strength products, qualitative research suggests that buying and consuming no- and low-alcohol beers are driven by health and wellbeing issues, price differentials, brand familiarity, improved product taste, and overall decreases in the social stigma associated with drinking alcohol-free beverages [49,50].

#### 3.2.2. What Factors Influence the Production of Lower Strength Products?

##### Grey Literature

One grey-literature report found that capital investments required for machinery to de-alcoholise beers, wines and spirits may be beyond the resources of smaller enterprises, requiring them to outsource de-alcoholization at costs per volume treated [18]. The same report found that increased production costs of de-alcoholization are offset by increased revenue streams that result from lower-alcohol strength products for both producers and vendors [18].

##### Academic Publications

Within the European Union, complimentary to working towards a stronger Health Union is a commitment to implementing the UN 2030 Agenda for Sustainable Development, in particular, its goals that address good health and wellbeing, clean water and sanitation, responsible consumption, and climate action, all of which are impacted by, and impact on, alcohol production [142]. For example, the water footprint is estimated to be 300 litres per one litre of beer and 870 litres per one litre of wine [51], and, for whiskey, for example, measured as a water scarcity footprint [52], of 790 litres per litre of 100% alcohol [53]. The carbon-dioxide equivalent (CO_2-eq_) emissions are approximately 0.575 to 0.842 kg per litre of beer (dependent on the packaging) [54], 0.85 kg per 70 cl bottle of wine [55] and, for whiskey, 4.4 kg per litre of pure alcohol [53].

Life cycle assessments (LCAs) for beer production suggest that onsite beer production is responsible for some 10% to 12% of CO_2-eq_ emissions; the biggest contributors of CO_2-eq_ emissions are cultivation and packaging [56,57,58,59]. LCAs for wine production suggest that most of the carbon footprint comes from cultivation and packaging [55,61,62]. LCAs for spirits production suggest that most of the carbon footprint comes from bottle production and packaging and onsite energy use [63]. One study from Sweden indicated that CO_2-eq_ emissions were lower for beers with an ABV ≤ 3.5% than for beers with an ABV >3.5%, but this difference was due to reduced transport-based emissions, with all lower alcohol-strength beers produced in Sweden, which was not the case for the higher strength beers. Insufficient information was analysed and presented to compare the differences in onsite beer production of CO_2-eq_ emissions between lower and higher strength beers [64]. No studies were found that compared LCAs between alcohol-free and regular-strength wine products. Priced-based interventions and interventions that change the assortment of products available to consumers can reduce the consumption of alcohol and can help control the environmental harms associated with their production, processing, transport and sale [65].

Due to global heating [66,67], the juice obtained from grapes at full phenolic maturation has an excessive concentration of sugar, resulting in wines with, at least form a health perspective, undesirably higher concentrations of alcohol, and this is likely to increase over time [68,69,70]. The effect can be mitigated through viticulture and winery tools to produce lower alcohol wines [71,72,73]. Late-ripening clones can be grafted onto the same variety, so that the wine typicity will not significantly change and the fruit ripening process can be delayed in order to cope with advanced phenology under rising temperatures [74]. From a long-term perspective, fruit ripeness can be considerably delayed by introducing late-ripening varieties to some important winemaking regions (e.g., Bordeaux) [75].

#### 3.2.3. Policies That Promote Lower Strength Products

##### Grey Literature

Two grey-literature publications found: (i) a modelling study in the United Kingdom estimated that, with a fixed duty per gram of alcohol that doubled with an ABV of between 2.0% and 5.0% and then doubled again between >5.0% and 40%, there would be an additional reduction in overall alcohol consumption of 5.4% compared to the present tax regime, with reductions in alcohol consumption due to off-trade (from shops, supermarkets etc.) beer of 0.2%, of off-trade wine of 4.7% and of off-trade spirits of 9.9% [76]. Taxes on the dose of alcohol rather than on the volume of the beverage may incentivize producers to reduce the volume of alcohol in beverages, [77] as seems to have been the case [78]; (ii) another modelling study across OECD countries found that alcohol policies in general, including setting a minimum price per gram of alcohol sold (which favours substitution), and regulating or banning advertising (which avoids potential pitfalls of substitution) reduce alcohol consumption and the harm done by alcohol, generate savings in heath expenditure, and improve employment and productivity, with high returns on investment [18]. For every EUR 1 invested in implementing a minimum price per gram of alcohol sold and a statutory ban on alcohol advertising targeting children, there could be EUR 13 annual economic benefit from lower rates of absenteeism, presenteeism and early retirement, and higher employment [18].

##### Academic Publications

The main policies that favour substitution of higher strength alcohol products with lower strength products relate to price, availability and improved labelling.

Household purchase data from Great Britain for the years 2015 to 2018 and for the first half of 2020 found that price promotions and lower prices increased household purchases of no- and low-alcohol beers [79]. Household purchase data from Great Britain for the years 2015 to 2018 and for the first half of 2020 found that, in relative terms, the ABV of purchased beer decreased by 2% following the introductions of a minimum unit price in both Scotland and Wales [79]. Household purchase data from Great Britain for the years 2015 to 2020 found that the proportion of all beers purchased with an ABV ≤ 3.5% increased in relative terms by 11% following the introductions of a minimum unit price in Scotland [80]. When looking at purchases, the introduction of minimum unit prices (MUPs) in Scotland and Wales, which promote substitution from higher to lower strength products [79,80], were associated with reductions in overall purchases of alcohol that were largely restricted to households that bought the most alcohol; the introduction of MUPs was not associated with an increased expenditure on alcohol by lower purchasing households, and, in particular, those with lower incomes [81,82]. In other words, the introduction of MUP did not appear to widen health inequalities. However, when looking at consumption, the introduction of a minimum unit price (MUP) in Scotland was not associated with reductions in consumption amongst younger men, men living in more deprived areas, and the top 5% of heaviest drinking men [83], for whom greater policy attention needs to be addressed.

Analyses in Saskatchewan in Canada found a 26% shift in sales of beer from higher to lower strength following increasing and setting slightly higher rates of minimum unit price according to five categories of beer strength [84].

Experimental bar studies suggest that increased availability of no- and low- alcohol products is associated with their increased selection and purchasing at the expense of higher strength products [85,86].

Experimental studies suggest that numerical descriptors on the label of low-alcohol beverages (e.g., %ABV) can lead to greater consumption of such products than verbal descriptors alone [87] (e.g., super low) in a sample of wine drinkers [88].

### 3.3. Mechanisms of Impact

#### 3.3.1. Do Current Consumers Substitute Higher Strength with Lower Strength Products? Grey Literature

Two grey-literature reports found that two fifths of respondents who bought and drunk no- and low-alcohol reported doing so to substitute higher strength products with lower strength products, often as a desire to drink less alcohol; whereas, one third reported that they had been using these products on top of, rather than instead of, existing levels of alcohol consumption [27,46].

##### Academic Publications

One publication analysing British household purchases of alcohol over the years 2015 to 2019 indicated substitution. Households that had previously bought same-branded regular-strength beers and who went on to buy newly introduced same-branded no- and low-alcohol beers subsequently reduced purchases of the regular-strength beers by 48.5 mL per adult per household per day for days in which a purchase was made, a 22.5% reduction, matched by new purchases of 34.6 mL of the new no- and low- alcohol beers, with such changes stable over at least two years follow-up (the length of time available for analyses) [89].

For Spanish households newly purchasing no-alcohol beer (ABV = 0.5%), the start of new purchases of no-alcohol beer, was associated with purchases of 116 mL of no-alcohol beer and decreases of 73 mL for all other with such changes remaining stable over four years, the length of analysis time [31].

For Spanish households newly purchasing no-alcohol wine (ABV = 0.5%), the start of new purchases of no-alcohol wine, was associated with purchases of 77 mL of no-alcohol wine and decreases of 92 mL for all other wines, with such changes remaining stable over four years, the length of analysis time [31].

#### 3.3.2. Does Buying and Drinking Lower Strength Products Act as a Gateway to Buying and Drinking Higher Strength Products?

The gateway hypothesis is predicated on a sequence of drug-use initiation with drug use itself viewed as the cause of drug-use development, with a progressive and hierarchical sequence of stages of drug use that begins with tobacco or alcohol, proceeds to marijuana, and from marijuana to other drugs, such as cocaine, methamphetamines and heroin [90]. An explanation for the development of involvement with psychoactive substances, however, seems better explained by a common liability to the use of psychoactive substances, which is grounded in genetic theory and supported by data identifying common sources of variation in the use of psychoactive substances, with identifiable neurobiological substrate and plausible evolutionary explanations [91].

##### Grey Literature

One grey-literature report found that, among Dutch school pupils aged 12 to 16 years, 9% reported drinking an alcohol-free drink at least once a week (12% of boys and 7% of girls), mainly non-alcoholic beer; those pupils who had drunk alcohol in the past month consumed non-alcoholic alternatives more often (16%) than those who had not drunk alcohol in the past month (7%) [143].

##### Academic Publications

For adults, household purchase data from Great Britain for the years 2015 to 2018 found that households that had never previously bought a same-branded higher strength beer but bought a new same-branded no- or low-alcohol beer were less than one third as likely to go on and newly buy the same-branded higher strength product as households that had never bought a new same-branded no- or low-alcohol beer, suggesting that no- or low-alcohol beverages did not act as triggers for higher strength products [89]. Household purchase data for both Great Britain and Spain indicate that, since the time of new purchases of no-alcohol products, substitution remained stable with no drift back to higher strength products for the length of time that it was studied (up to two years in Great Britain [89] and up to four years in Spain [31]).

For adolescents, during all of the 2000s, there has been a steady decline in alcohol consumption amongst 12–17 year olds throughout all of the EU, and in most high-income countries [144]. Such a decline has taken place at the same time as a general increase in the availability of no- and low-alcohol products [27], which might counter an argument that no- and low-alcohol products act as predominant triggers to consumption of higher strength products by young people. The realist review found one study of young people to test the idea: a Japanese study of over 100,000 adolescents reported that the use of alcohol-flavoured non-alcoholic beverages (AFNAB) usually started after adolescents began consuming alcohol, and not the other way round [92].

Although some surveys find that one in fourteen consumers of no/low products report doing so because they are recovering from “alcohol dependence” [46], one study found no differences in brain activation between beer and non-alcoholic-beer tasting, in particular not in brain areas involved in reward receipt, suggesting that in regular consumers, beer flavour rather than the presence of alcohol could be an important driver of the consumption experience [145]. Thus, the taste of non-alcoholic beers could act as relapse triggers [93] for alcohol dependence [94,95], particularly those with high sugar content [96]; no studies were identified by the realist review that investigated this.

#### 3.3.3. Is There Additional and Alibi Marketing Due to the Introduction of Lower Strength Alcohol Products?

##### Grey Literature

Five grey-literature publications found or stated: (i) marketing strategies by producer companies to promote no- and low- alcohol products include opening up new contexts and times to drink (addition marketing), selling lifestyles and identities, and sports marketing/sponsorship [49]; (ii) a Dutch survey of 15–17 year olds and of adults found that, whilst 15% agreed with the statement that marketing of no-alcohol products would normalize drinking alcoholic beverages, almost 60% thought that it could encourage drinkers to drink alcohol-free alternatives instead of alcoholic beverages [97]; (iii) qualitative interviews find that consumers of no- and low-alcohol beverages report resistance to addition marketing, wanting instead to substitute higher strength products for no- and low-alcohol products [49]; (iv) it is proposed that a backdrop of effective marketing regulation [49] needs to be in place to avoid the use of alcohol-free and no-alcohol products circumventing existing marketing regulations for same-branded higher strength alcoholic beverages (alibi marketing [98,99]), and the targeted marketing of new consumer groups or new drinking occasions, as emphasized by WHO [19]; and, (v) a backdrop of effective marketing regulation can be integrated within element ‘E’ of WHO’s SAFER initiative, “Enforce bans or comprehensive restrictions on alcohol advertising, sponsorship, and promotion” [17], irrespective of the ABV of the product down to 0.0%, as implemented in some countries [100].

##### Academic Publications

In a context of an extensive volume of alcohol marketing [101], a range of reviews have concluded a causal connection between alcohol marketing in a range of media and young people’s alcohol consumption [102,103,104,105,106,107,108,109]. Whilst no specific analyses of the impact of advertising of zero- and no-alcohol beverages on youth alcohol consumption have been identified, in general, alcohol brands with youth-appealing advertising are consumed more often by youth than adults, indicating that these advertisements may be more persuasive to relatively younger audiences [110].

With respect to marketing, household purchase data from Great Britain for the years 2015 to 2018 found that households were more than twice as likely to buy a newly introduced no- or low-alcohol beer if they had previously bought the same-branded higher strength beer [89].

#### 3.3.4. Is There Policy Interference Following the Introduction of Lower Strength Products?

##### Grey Literature

Concern has been expressed that no- and low- alcohol products may broaden the normalization of drinking cultures, including in environments where drinking does not normally take place, such as in the workplace [34], or may counter the de-normalization of drinking that is currently occurring, at least amongst European youth [144].

##### Academic Publications

Substitution of higher strength products with lower strength products are an addition to, and not a replacement of, the mainstay of reducing the harm done by alcohol, which is the strengthened enforcement of the high-impact cost-effective policies included in WHO’s SAFER technical package [17,19]. Governments need to ensure that, as alcohol producers take responsibilities for their products by substituting higher- with lower strength products, alcohol producers take responsibility for not further encroaching the policy environment as they [111,112,113,114,115], like the food industry [116], do. WHO calls on economic operators in alcohol production and trade “to abstain from interfering with alcohol policy development and refrain from activities that might prevent, delay or stop the development, enactment, implementation and enforcement of high-impact strategies and interventions to reduce the harmful use of alcohol.” [19].

### 3.4. Outcomes

#### 3.4.1. Does Substitution Recue Alcohol Consumption?

##### Grey Literature

None.

##### Academic Publications

In Great Britain during 2015–2018, based on 3.2 million separate alcohol purchases by 64,286 households, interrupted time-series analyses found that the introduction of 46 new no- and low- (ABV ≤ 3.5%) alcohol beers and the reformulation of 33 existing beers to contain less alcohol (out of 1903 available beer brands) was associated with reductions in purchases of all grams of alcohol across all households, larger for reformulation (3.9%) than for the introduction of new no- and low-alcohol beer (2.6%); and, were larger for households that bought the most alcohol [29].

In Great Britain during 2015–2019, based on 4 million separate alcohol purchases by 69,803 households, time-series analysis found that for every 10 mL increase in purchases of alcohol-free beer per adult per household per day (from a baseline of 10 mL), purchases of grams of all alcohol contained within beer dropped by 1.1%; for every 5 mL increase in purchases of alcohol-free wine products per adult per household per day (from a baseline of 5 mL), purchases of grams of all alcohol contained within wine dropped by 1.2% [30].

In Great Britain during 2015–2019, based on 4 million separate alcohol purchases by 69,803 households, the ABV of beer decreased over time; time-series analysis found that, for every drop in the absolute value of ABV of 0.1% over time (from a baseline of 4.34), the associated drop in purchases of grams of all alcohol contained within beer was 6.9% [30].

For Spanish households newly purchasing no-alcohol beer, the start of new purchases of no-alcohol beer was associated with decreased purchases of all other beer and a drop in purchases of grams of all alcohol of 5.3 g (95% CI = 5.0 to 5.7) per adult per household per day of purchase, a 5% drop which remained stable over the full four years of follow-up, with reductions in purchased grams of alcohol greater the higher the volume of all other beer purchases prior to the new purchases of no-alcohol beer [31].

For Spanish households newly purchasing no-alcohol wine, the start of new purchases of no-alcohol wine was associated with decreased purchases of all other wine and a drop in purchases of grams of all alcohol of 8.2 g (95% CI = 7.8 to 8.6) per adult per household per day of purchase, an 8% drop which remained stable over the full four years of follow-up, with reductions in purchased grams of alcohol greater the higher the volume of all other wine purchases prior to the new purchases of no-alcohol wine [31].

In Spain, at the beginning of 2021, two new same-branded 20% ABV variants of whiskey and gin were launched. Households that purchased the 20% variants did not switch purchases from the same-branded regular-strength products, but did switch purchases from other spirits products to the 20% variants, with reductions in purchases of 26.7 g of alcohol in all spirits products (95% CI = 23.6 to 29.8) per adult per household per purchase day, a 17% drop, with the mean % ABV of purchased spirits decreasing by 6.1 (95% CI = 5.8 to 6.4), a 15% drop [31].

#### 3.4.2. Does Substitution Improve Health?

##### Grey Literature

None.

##### Academic Publications

The realist review found no publications that reported empirical analyses of health outcomes of consumers who substituted higher strength products with lower strength products. In the Northern Territory of Australia, one study reported time-series analyses that found that a tax levy on beverages with an ABV >3.0%, combined with community-based programmes, was associated with reductions in alcohol-attributable deaths [117]. It was not possible, though, to fully separate the independent effects of the tax levy (which was used to finance the community-based programmes) and the community-based programmes themselves.

In a modelling scenario where, in each country of the WHO European Region (with 753.5 million inhabitants older than 15 years of age), the final price per gram of alcohol is the same for all alcohol products (equalization), with that final price based on a minimum tax share of 15% of the retail price of the beverage in each country that has the highest price per gram of alcohol independent of taxation, then 133,000 deaths could be averted each year [146]. (More than three times as many deaths, 40,000, that could be averted were only a minimum tax share of 25% implemented, with no equalization.)

Recent reviews have attempted to summarize the evidence of potential physiological health benefits from components of no- and low-alcohol products other than alcohol [147,148,149,150,151,152,153], including phytoestrogens [154,155], inflammasomes [118,119], microRNAs [120,121,122], and polyphenols [123,124,125,126,127,128,151]. Small randomised-controlled trials have demonstrated that no-alcohol beers and wines either had improved or same potentially beneficial physiological health outcomes due to the components as regular strength beers and wines (e.g., [129,130,131,154,155]), all in the absence of the toxic effects of alcohol [132]. The extent to which these potential benefits compare to the health benefits of reduced alcohol consumption has not been studied, although they are likely to be marginal.

In general, non-alcoholic beers have a lower energetic value, but higher sugar content than alcoholic beers [124,133,134].

The alcohol in products with an ABV ≤ 0.5% would be metabolized by the first pass metabolism of the stomach and liver [135,136,137], and are, likely, of no measurable health risk, [16] being below the safety margin of 2.6 g alcohol per day, based on margins of exposure analyses [138].

## 4. Discussion

The realist review found that, albeit from a limited number of existing studies, drinkers were substituting higher strength alcohol products with lower strength products, often doing so to buy and drink fewer grams of alcohol. Such substitution resulted in fewer grams of alcohol bought and drunk, which, although not specifically studied, would, at the individual level, lead to health gain. Although all segments of the population were engaged in substitution, it was led by more social-economically advantaged younger men, and by existing heavier buyers and drinkers of alcohol, in the context of samples which, by their nature, tend not to represent the very heaviest drinkers in the population. Whilst a few small experimental studies demonstrated that other constituents of no-alcohol beverages (with the alcohol removed) could have health advantages, the size of these advantages vis a vis the lower risk of ill health experienced by removing alcohol has not been documented but is likely to be marginal. Other potential health gains will depend on the counterfactual; for example, substitution occurring amongst drivers and amongst women who are pregnant or breast-feeding would result in less harms to others than the drinker.

Within the European context, alcohol producers seem to be playing catch up and responding to consumer demand by increasing the availability of lower strength products, but market penetration remains very low and insufficient to have noticeable impact on health improvement at the population level, despite health improvement likely to occur at the individual level. Public-health policy making seems to lag behind consumer and producer changes, with little action to favour or manage substitution, and large missed opportunities to favour substitution through lack of adjustment of existing administrative and policy measures.

The theory of change and logic model examined in this realist review are not unique to alcohol. Within the context of promoting healthier eating, for example, substitution and product reformulation are mainstays of change for, amongst others, salt, sugar and trans fats, all of which improve health [139,140,156,157,158,159,160,161,162,163]. Notwithstanding that alcohol is a toxic and psychoactive substance [164], one can still learn from these initiatives. Take meat, for example: we all have to eat less meat to reduce planet-heating greenhouse gas emissions [66] (and, improve health [165,166])—in making that shift, consumers crave familiarity and are influenced by the same physical and linguistic cues—they want the same burger without the meat [167,168]; for alcohol, they want the same beer, with the same look, the same taste, the same price promotion, and the same supermarket shelf, but without the alcohol [50].

As performed by this paper, a strength of the realist review is that the approach is pluralist and flexible, summarizing findings from a wide range of qualitative and quantitative research methods reported in published grey and academic literature, driven by the aim of better understanding what it is about lower strength alcohol products that could reduce risk to ill-health, for whom and in what circumstances, and in what respects and why [20,21]. By taking a pluralist and flexible approach, the realist review has learnt from, rather than controlled for, real-world phenomena and experience. By taking better understanding of lower strength alcohol products as the core of analysis, the realist review has maximized learning from the available evidence across policy, disciplinary and sectoral boundaries.

As performed by this paper, the realist review also has important shortcomings. The realist review is not a protocol-driven approach, but more about identifying both understanding and principles that can potentially guide policy with respect to lower strength alcohol products. The trails of our literature searches, the interpretation of the findings, and quality assurance of searches and identified papers are based on our own judgements and are, thus, not necessarily standardizable or reproducible in the same sense as, say, a conventional Cochrane systematic review, in which key quality features are technical standardization and clarity of presentation. The findings from our realist review are more like a road map for substituting higher strength alcohol products with lower strength products, alerting policymakers to the problems that might arise and how to deal with them, rather than producing generalizable effect sizes of specific interventions.

A further shortcoming of our paper is the limited availability of evidence and that much of the evidence is restricted to a small number of data sources and jurisdictions, leading to concerns about the applicability of the evidence to different countries, and cultural and drinking contexts. A large investment in further research and monitoring is necessary to see how no- and low-alcohol products can fit within the policy and drinking context of other countries, and especially to monitor if the alcohol industry is tailoring approaches to countries in how they are using promotion of no- and low-alcohol products. Ongoing research is needed to monitor the impact of the increasing availability and consumption of no- and low-alcohol products on consumer behaviour and on public health, including unintended consequences, with findings from research informing governmental policies at all levels.

## 5. Conclusions and Road Map for Alcohol Policy

WHO’s call on economic operators, to “substitute, whenever possible, higher-alcohol products with no-alcohol and lower-alcohol products in their overall product portfolios, with the goal of decreasing the overall levels of alcohol consumption in populations and consumer groups,” [19] appears, at least from the perspective of consumers, to be evidence-based. Although limited, the existing evidence demonstrates that substitution results in fewer grams of alcohol purchased.

Alcohol policy that favours substitution is one that facilitates increased availability of lower strength products, provides such products with clearer labelling, and sets alcohol taxes that increase based on the (mathematical) product of the ABV. A disproportionate uptake of substitution by more affluent consumers needs to be balanced by social-norm campaigns [169] that extend the reach of lower strength products to all of society, and implementation of evidence-based alcohol policy measures that lessen health inequalities [170]. In Europe, at least, the full potential of alcohol policy is held back by a range of administrative regulations that hamper increased availability and improved labelling and by tax requirements that, if anything, favour higher strength products [171,172].

WHO has set out clear guidance to manage problems that might occur with substitution: (i) whilst, within individual brands, substitution occurs from higher to lower strength products rather than the other way around, a backdrop of effective marketing regulation [49] needs to avoid the use of alcohol-free and no-alcohol products circumventing existing marketing regulations for same-branded higher strength alcoholic beverages (alibi marketing [98,99]), and the targeted marketing of new consumer groups or new drinking occasions, as emphasized by WHO [19]. A backdrop of effective marketing regulation can be integrated within element ‘E’ of WHO’s SAFER initiative, “Enforce bans or comprehensive restrictions on alcohol advertising, sponsorship, and promotion” [17], irrespective of the ABV of the product down to 0.0%, as implemented in some countries [100]; (ii) greater responsibility in the policy environment, as called for by WHO, requires producer companies to abstain from “interfer(ing) with alcohol policy development and refrain(ing) from activities that might prevent, delay or stop the development, enactment, implementation and enforcement of high-impact strategies and interventions to reduce the harmful use of alcohol.” [19].

Governments, themselves, need to increase their responsibilities through strengthened implementation and enforcement of the high-impact cost-effective policy options itemised in WHO’s SAFER technical package [17], and amend the administrative restrictions to effective alcohol policy that they, themselves, have made, and which lead to avoidable lost lives.

## Figures and Tables

**Figure 1 nutrients-14-03779-f001:**
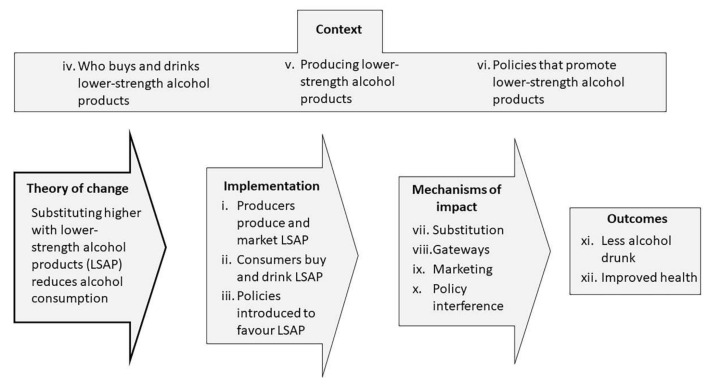
Logic model in which the review examines questions related to implementation, uptake of lower strength alcohol products that takes place at the levels of (i) producers, (ii) consumers, and (iii) policy-makers; within a context of (iv) who buys and drinks lower strength products and why, (v) the production of lower strength products, and (vi) policies that promote the uptake of lower strength products; with mechanisms of impact that are potentially beneficial through (vii) substitution, and potentially adverse with lower strength alcohol products acting as (viii) gateways to higher strength products, (ix) mediators of additional and alibi marketing, and (x) facilitators of increased interference by alcohol producers in alcohol policy; leading to outcomes of (xi) less alcohol drunk and (xii) improved health.

**Table 1 nutrients-14-03779-t001:** Summary of evidence from realist review based on logic model of Figure 1.

Framework Category	Question	Key Findings	Evidence Limitation	Associated References
Implementation	What is the extent of the production of lower strength alcohol products?	Production is low with large differences between European countries, largely restricted to beer, but increasing.	Routine reported production restricted to beer.	[27,28,29]
	To what extent are consumers buying and drinking lower strength alcohol products?	Purchase and consumption is low with large differences between European countries, largely restricted to beer, but increasing.	Very little routine data collected; household purchase data restricted to two countries (Great Britain and Spain).	[28,30,31]
	What are the currently implemented policies relevant for lower strength alcohol products?	Despite some intentions, little policy is set at country or European levels that might favour substitution. On the other hand, many existing policies set at country and European levels disfavour substitution.	What policy set at European level to favour substitution has not been evaluated.	[32,33,34,35,36,37,38,39,40,41,42,43]
Context	Who buys and drinks lower strength alcohol products and why?	In general, it seems, at least for beer, that younger and those with higher incomes are more likely to buy and drink no- and low-alcohol products, in about two-fifths of cases report doing so to drink less alcohol.	Mostly based on grey, rather than academic literature.	[27,44,45,46,47,48,49,50]
	What are factors influencing the production of lower strength products?	Increased production costs of de-alcoholization offset by increased revenues. Global heating leads to higher strength wines. Life cycle assessments suggest increased global warming potential of de-alcoholization likely to be marginal, as most global warming potential comes from cultivation and packaging.	Insufficient information available on life cycle assessments of de-alcoholization.	[51,52,53,54,55,56,57,58,59,60,61,62,63,64,65,66,67,68,69,70,71,72,73,74,75,76,77]
	What policies should be set that can gain health benefits from lower strength alcohol products, whilst avoiding the negative consequences?	Modelling studies suggest that taxes that rise with alcohol by volume (ABV) steeper at the lowest ABV levels would favour substitution. Empirical analyses of household purchase data find that the introduction of minimum unit price favours substitution. Experimental studies suggest that increased availability and improved labelling of no- and low-alcohol products would favour substitution.	The findings from minimum unit price are robust, but overall, evidence base for policy limited.	[78,79,80,81,82,83,84,85,86,87,88,89,90]
Mechanisms of impact	Do consumers substitute higher strength with lower strength products?	Household purchase data from Great Britain and Spain indicate substitution for purchases of beers and wines.	Scientific publications limited to household purchase data from two jurisdictions.	[27,31,46,91]
	Does buying and drinking lower strength products act as a gateway to buying and drinking higher strength products?	Available evidence from youth surveys and household purchase data suggests not. Impact of no- and low- alcohol products for those with a diagnosis of “alcohol dependence” unknown.	Youth survey data limited to one grey-literature Dutch study and one academic Japanese study. Publications based on household purchase data limited to Great Britain and Spain.	[31,91,92,93,94,95,96,97,98]
	Is there additional and alibi marketing due to introduction of lower strength alcohol products?	Additional and alibi marketing appears to exist; despite considerable evidence of the impact of advertising on youth consumption, no specific studies of impact of no- and low-alcohol products on consumption behaviour. At least for beer, brand loyalty seems to favour switching from higher to lower strength products, but not the other way round.	No specific evidence available on impact of no-alcohol advertising on youth behaviour. Publications based on household purchase data limited to Great Britain	[49,91,99,100,101,102,103,104,105,106,107,108,109,110,111,112]
	Is there additional policy interference because of focus on lower strength alcohol products?	Whilst alcohol producers do interfere with the policy environment, no documented studies describe interference related to no- and low-alcohol products.	No documented studies that describe specific interference related to no- and low-alcohol products.	[113,114,115,116,117]
Outcome	Does substitution reduce alcohol consumption?	Household purchase data from Great Britain and Span demonstrate that substitution is associated with decreased purchases of grams of alcohol overall, in relation to beers, wines and spirits.	Publications based on household purchase data limited to Great Britain and Spain.	[29,30,31]
	Does substitution improve health?	Empirical analyses of health outcomes subsequent to substitution have not been identified; however, as substitution results in fewer grams of alcohol at least purchased, and reduced consumption results in health gains, substitution likely to improve health. Small randomised-controlled trials have demonstrated that no-alcohol beers and wines either had improved or same potentially beneficial health outcomes as regular strength beers and wines from other components in the absence of the toxic effects of alcohol; however, the extent to which these potential benefits compare to the health benefits of reduced alcohol consumption has not been studied, although likely to be marginal. In general, no-alcohol beers have lower energetic value but higher sugar content than alcoholic beers.Alcohol content of no-alcohol products likely to be of no measurable health risk.	No empirical analyses available.	[118,119,120,121,122,123,124,125,126,127,128,129,130,131,132,133,134,135,136,137,138,139,140]

## Data Availability

No additional data available.

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
