# Peer review of "Lower Strength Alcohol Products—A Realist Review-Based Road Map for European Policy Making"

_nutrients, 2022, doi:10.3390/nu14183779_

Round 1

Reviewer 1 Report

This work addresses a proposed logic model based on Medical Research Council’s guidance related with lower-strength alcohol products. The authors examine each of the twelve components of the logic model in the light of the literature that came out between since 1st January 2011 until 31 May 2022.

The theme is well founded through a consistent literature review. The objectives are correctly defined. The methodology is well-designed and is consistent with the objectives of the study. The interpretation and discussion of results is clear, objective, and consistent. The conclusions summarize well the results obtained and are consistent with the work presented.
However, there are some important notes that should be considered:

 1)   The abstract should include a brief account of the work developed by the authors, the methodology used, and the results obtained, and not just the framing of the theme.

2) It is important to review all document formatting. The different sections must be properly numbered.

Author Response

Reviewer 1

The abstract should include a brief account of the work developed by the authors, the methodology used, and the results obtained, and not just the framing of the theme.

RESPONSE: Thank you for your comments. We have revised the abstract.

It is important to review all document formatting. The different sections must be properly numbered.

RESPONSE: We have numbered the sections.

Reviewer 2 Report

Anderson et al. submitted a “realist” review of information on how promotion of reduced- or non-alcoholic beverages can affect overall intake of ethanol. While this is an interesting manuscript, this reviewer does not believe that Nutrients would be an appropriate journal for such a paper. It is suggested that it be considered for publication in a journal related to public policy or social sciences. Still, I do have a few comments below that may be helpful for the authors to consider:  

ABSTRACT

-        Line 17. “health gain” – would reword to indicate lower risk for diseases related to alcohol consumption. The term health gain is too broad.   

INTRODUCTION

-        Line 36. The authors mention “ethanol” (etOH) but then switch to using the term alcohol. Please clarify whether alcohol is being used to refer to alcoholic beverages or to ethanol alone. Please keep consistent throughout the manuscript

-        Line 38. Please double check some of the references that are being cited. For instance, one of the references cited here is specifically referring to the ethyl carbamate in alcoholic beverages, not to etOH as referred to in the statement. The authors here should make a distinction between the etOH itself and the alcoholic beverages. Both are important aspects to consider, but it is important to note that a compound such as ethyl carbamate is not necessarily correlated to the % alcohol of a beverage.     

Author Response

Reviewer 2

This is an interesting manuscript, this reviewer does not believe that Nutrients would be an appropriate journal for such a paper. It is suggested that it be considered for publication in a journal related to public policy or social sciences.

RESPONSE: Thank you for your comments.  This is a paper for a special issue of Nutrients, termed “Lower Strength Alcohol Products and Public Health”. The first author of the paper is also the guest editor of the special issue.

ABSTRACT

-        Line 17. “health gain” – would reword to indicate lower risk for diseases related to alcohol consumption. The term health gain is too broad.   

RESPONSE: We have re-worded.

INTRODUCTION

-        Line 36. The authors mention “ethanol” (etOH) but then switch to using the term alcohol. Please clarify whether alcohol is being used to refer to alcoholic beverages or to ethanol alone. Please keep consistent throughout the manuscript

RESPONSE: Thank you for pointing this out.  As alcohol is a more commonly used term, we have stuck to the term alcohol throughout. 

-        Line 38. Please double check some of the references that are being cited. For instance, one of the references cited here is specifically referring to the ethyl carbamate in alcoholic beverages, not to etOH as referred to in the statement. The authors here should make a distinction between the etOH itself and the alcoholic beverages. Both are important aspects to consider, but it is important to note that a compound such as ethyl carbamate is not necessarily correlated to the % alcohol of a beverage.     

RESPONSE: The IARC report of reference 9 is in two separate parts. The first part assesses the carcinogenicity of alcohol consumption. The second part specifically assesses the carcinogenicity of ethyl carbamate (one of some 16 carcinogenic components of alcoholic beverages that have had separate reviews). So, as a reference for the carcinogenicity of alcohol consumption, it is correct.